# Egg Quality of Italian Local Chicken Breeds: II. Composition and Predictive Ability of VIS-Near-InfraRed Spectroscopy

**DOI:** 10.3390/ani13010077

**Published:** 2022-12-25

**Authors:** Filippo Cendron, Sarah Currò, Chiara Rizzi, Mauro Penasa, Martino Cassandro

**Affiliations:** 1Department of Agronomy, Food, Natural Resources, Animals and Environment (DAFNAE), University of Padova, Viale dell’Università 16, 35020 Legnaro, Italy; 2Department of Comparative Biomedicine and Food Science, University of Padova, Viale dell’Università 16, 35020 Legnaro, Italy; 3Federazione delle Associazioni Nazionali di Razza e Specie, Via XXIV Maggio 43, 00187 Roma, Italy

**Keywords:** chemical composition, hen, infrared technology, native breed, quality

## Abstract

**Simple Summary:**

Veneto is one of the most important Italian regions for poultry biodiversity. However, the indigenous population cannot compete, in terms of production, with the commercial lines. The nutritional value significantly varies among the breeds as a consequence of changes in egg composition and size. The variability in terms of egg composition is useful to develop Near-InfraRed spectroscopy calibration models for its prediction.

**Abstract:**

The aims of the present study were to characterize egg composition and develop VIS-Near-infrared spectroscopy (VIS-NIR) models for its predictions in Italian local chicken breeds, namely Padovana Camosciata, Padovana Dorata, Polverara Bianca, Polverara Nera, Pepoi, Ermellinata di Rovigo, Robusta Maculata and Robusta Lionata. Hens were reared in a single conservation center under the same environmental and management conditions. A total of 200 samples (25 samples per breed, two eggs/sample) were analyzed for the composition of albumen and yolk. Prediction models for these traits were developed on both fresh and freeze-dried samples. Eggs of Polverara Nera and Polverara Bianca differed from eggs of the other breeds (*p* < 0.05) in terms of the greatest moisture content (90.06 ± 1.23% and 89.57 ± 1.31%, respectively) and the lowest protein content (8.34 ± 1.27% and 8.81 ± 1.27%) in the albumen on wet basis. As regards the yolk, Robusta Maculata and Robusta Lionata differed (*p* < 0.05) from the other breeds, having lower protein content (15.62 ± 1.13% and 15.21 ± 0.63%, respectively) and greater lipid content (34.11 ± 1.12% and 35.30 ± 0.98%) on wet basis. Eggs of Pepoi had greater cholesterol content (1406.39 ± 82.34 mg/100 g) on wet basis compared with Padovana Camosciata, Polverara Bianca and Robusta Maculata (*p* < 0.05). Spectral data were collected in reflectance mode in the VIS-NIR range (400 to 2500 nm) using DS2500 (Foss, Hillerød, Denmark) on fresh and freeze-dried samples. Models were developed through partial least-squares regression on untreated and pre-treated spectra independently for yolk and albumen, and using several combinations of scattering corrections and mathematical treatments. The predictive ability of the models developed for each compound was evaluated through the coefficient of determination (R^2^cv), standard error of prediction (SEcv) and the ratio of performance to deviation (RPDcv) in cross-validation. Prediction models performed better for freeze-dried than fresh albumen and yolk. In particular, for the albumen the performance of models using freeze-dried eggs was excellent (R^2^cv ≥ 0.91), and for yolk it was suitable for the prediction of protein content and dry matter. Good performances of prediction were observed in yolk for dry matter (R^2^cv = 0.85), lipids and cholesterol (R^2^cv = 0.74). Overall, the results support the potential of infrared technology to predict the composition of eggs from local hens. Prediction models for proteins, dry matter and lipids of freeze-dried yolk could be used for labelling purposes to promote local breeds through the valorization of nutritional aspects.

## 1. Introduction

In the last sixty years, the selection of high-performing commercial chickens has hidden the productive characteristics of the local breeds, which are more disadvantageous from a commercial point of view due to the lower oviposition rate. This has had significant effects on the animal production scenario, leading to genetic erosion of local species and breeds through the loss of genetic variability [1,2]. In particular, genetic depletion has characterized the poultry sector, incentivizing production systems based on commercial hybrids in order to reach greater performances for both meat and egg production than the local breeds. Although local breeds are considered less profitable than commercial hybrids, the territoriality of these breeds is a very important resource, mirroring the culture and heritage of the local producers. Moreover, the heterogeneous genetic background of the autochthonous breeds affects the variability of the final product, conferring qualitative traits that differ among breeds.

As cheap and nutritional complete food, egg has been recognized as an essential product of the daily diet of humans, providing almost all the main elements of human nutrition, such as moisture (75%), lipids (12%), proteins (12%), carbohydrates (1.1%) and minerals (0.5%) [3]. Proteins are included, both in egg albumen and yolk, whereas lipids and cholesterol are exclusively found in the yolk. The characterization of egg composition is important to evaluate the nutritional profile on the commercial label. 

Local breeds, which have not undergone any genetic selection to improve egg production and quality, are not suitable for intensive farming, as their egg production satisfies only a niche market. They show slow growing rate, delayed onset of laying when compared to commercial hybrids and, overall, they reach adult body weight at 6 months of age or even later [4]. However, these breeds can be valorized for their meat production both for males and females, after their egg production cycle. Breeds laying white-shell eggs, especially Padovana, which shows a poor muscle growth, produce meat appreciated for the peculiar traits and used for typical cooking preparations [4]. Breeds laying tinted-shell eggs have body weight ranging from 1 kg (Pepoi) to 3.3 kg (Robusta Lionata), and the carcass shows an appreciable muscle development [5,6].

Generally, egg composition is determined through laboratory analyses which, despite having high accuracy, are time-consuming, require skilled personnel and sample destruction, and are not applicable on a large scale [7]. The development of bioinformatics approaches and emerging technologies has made spectral analysis an important technique in the food sector [7]. VIS-Near-InfraRed spectroscopy (VIS-NIR) is a technique with a wide range of applications in food, such as the determination of the composition of products. This technology is cheap, sample preparation is easy, and the analysis is non-destructive. In the last decade, the VIS-NIR has been widely used to determine egg quality [3], freshness [7], blood and meat spots [8] and egg-shell composition [9], and also to detect infertile and non-hatchable eggs [9]. Currently, the importance of eggs from hybrid hens has marginalized the potential interest that the local breeds could have. To the best of our knowledge, nobody has attempted to develop VIS-NIR models to predict the composition of eggs from local breeds. Therefore, this study aimed to characterize the egg composition of eight local chicken breeds of the Veneto region (Northern Italy) and to investigate the ability of VIS-NIR to predict the composition traits.

## 2. Materials and Methods

### 2.1. Hens and Farming Conditions 

Hens were located in the Conservation Centre “Sasse Rami” (Ceregnano, Italy) managed by Veneto Agricoltura Agency and were drawn from 8 local breeds, namely Padovana Camosciata (PA-C, chamois plumage), Padovana Dorata (PA-G, golden plumage), Polverara Bianca (PO-W, white plumage), Polverara Nera (PO-B, black plumage), Pepoi (PP), Ermellinata di Rovigo (ER), Robusta Maculata (RM) and Robusta Lionata (RL) (Figure 1). The first four breeds produce white-shell eggs, whereas the latter four breeds lay tinted-shell eggs. The PA-C, PA-G, PO-B and PO-W are recognized as egg-type breeds, whereas the other are dual-purpose (egg and meat) breeds.

Each breed was reared in a free-range area of about 300 m^2^ (5 m^2^/hen), equipped with linear drinkers, and hens had access to a wooden structure of 15 m^2^ (approximately 0.25 m^2^/hen) and 1 m^2^ of space occupied by the collective nest, with litter made of wood shavings. The wooden structure was equipped with perches, circular feeders and artificial light for complementing the natural photoperiod. The light cycle, according to the period of the year, was regulated by a timer to obtain an increasing photoperiod up to 14 h of light at the time of eggs collection, which was achieved by turning on the light one hour before sunrise and turning off it one hour after sunset. The hens were free to enter and leave the wooden structure, which they used mainly during the night, feeding, laying and in case of bad weather.

The diet was composed of commercial feed, consisting mainly of maize and soybean. The hens were fed ad libitum, starting from the laying period using pelleted feed (composition, % as-fed basis: crude protein = 16.2, Ca = 4.2, *p* = 0.6, lysine = 0.7, methionine = 0.3). The metabolizable energy was 11.5 MJ/kg. Feeding, rearing conditions (temperature, photoperiod) and prophylaxis procedures were the same for all breeds from the time of hatching until the end of the testing period.

### 2.2. Sampling and Chemical Analysis of the Eggs

The eggs were collected according to the European Regulations (EC No. 1/2005 and EC No. 1099/2009) on animal care and welfare. The sampling did not affect the welfare of the hens as it was carried out when the animals were not in the nests, thus avoiding their handling. For this study, 60 hens per breed were sampled. Hens averaged 50 weeks of age (with very small variation, from 49 to 51 weeks of age). Fifty eggs per breed across 4 consecutive days (about 13 eggs/breed/day) were collected, which resulted in a total of 400 eggs collected during the trial and 200 final samples (25 samples/breed) used for composition analyses, since each sample was composed of two eggs. Indeed, albumens and yolks of eggs collected during the trial were freeze-dried, and albumen (pool of albumens from two eggs) and yolk samples (pool of yolks from two eggs) were separately frozen at −20 °C. Samples were analyzed for moisture, proteins, lipids and ash according to the official AOAC analytical methods. Albumen and yolk dry matter [AOAC 925.30], proteins [AOAC 925.31], ash [AOAC 920.153] and total lipid contents [AOAC 991.31] were determined [10]. For cholesterol determination, tubes containing freeze-dried yolk samples (100 mg) were added with 5 mL of ethanol (95%) (Sigma-Aldrich, Burlington, NJ, USA) and 2 mL of KOH (Sigma-Aldrich, Burlington, NJ, USA), placed in a water bath at 70 °C and shaken for 10 min. After cooling at room temperature, 1 mL of internal standard (pregnenolone-ethanol; Sigma-Aldrich, Burlington, NJ, USA) and 35 mL of hexane-ethyl ether (Sigma-Aldrich, Burlington, NJ, USA) were added. After the addition of 20 mL of deionized water, samples were centrifugated. An aliquot of the supernatant organic phase (25 mL) was taken and dried in a rotating evaporator kept in a water bath at 35 °C (ASAL, Milano, IT). Then, 5 mL of the mobile phase was analyzed by GC-FID (Agilent, Santa Clara, CA, USA) [11].

### 2.3. Near-InfraRed Spectra Collection

Individual spectra were collected on the albumen and the yolk separately, both on fresh and freeze-dried samples. Each part of the egg was placed in a large sample cup (diameter 105 mm, depth 35 mm) and scanned with NIRS DS2500 (FOSS Electric A/S, Hillerød, Denmark) every 0.5 nm, from 400 to 2500 nm wavelength, at room temperature. Spectra were collected through ISIscan Nova and Mosaic software (FOSS Electric A/S, Hillerød, Denmark) and recorded as log(1/reflectance).

### 2.4. Chemometric Data Analysis

Spectral chemometric analysis was performed using WinISI software (Infrasoft International, Port Matilda, PA, USA). Four different prediction equations were developed for albumen and yolk, for both fresh and freeze-dried samples, using the modified partial least-squares (PLS) regression. To avoid overfitting, the PLS regressions were performed with five cross-validation steps and the number of PLS terms was limited to a maximum of ten. Spectral samples with a predicted value that differed more than 2.5 standard deviations from the reference value were excluded from the dataset (outliers). Several combinations of scatter corrections (NONE, no correction; SNV, standard normal variate; SNV_D, standard normal variate-detrending; MSC, multiplicative scatter correction) and derivative mathematical treatments (0, 0, 1, 1; 1, 4, 4, 1; 2, 5, 5, 1; where the first digit is the number of the derivative, the second is the gap over which the derivative is calculated, the third is the number of data points in the first smoothing and the fourth is the number of data points in the second smoothing) were tested. The performances of the prediction models were evaluated through the standard error of calibration (SE_C_) and cross-validation (SE_CV_); the coefficient of determination of calibration (R^2^_C_) and cross-validation (R^2^_CV_); and the residual predictive deviation of cross-validation (RPD_CV_), calculated as the ratio between SD and SE_CV_ [12].

### 2.5. Statistical Analysis and Data Visualization

Normal distribution of the investigated traits was assessed through Shapiro–Wilk test and visual inspection of the normal plot. All traits were normally distributed and thus were analyzed through one-way analysis of variance, considering the effect of the breed. The same statistical approach was used to test differences between the composition of white versus tinted eggshell breeds. A multiple comparison of means was performed for the effect of breed, using Tukey’s post hoc test. Significance was set at *p* < 0.05, unless otherwise stated. All statistical analyses were carried out using R software version 4.04 [13]. Data were visualized by R package *ggplot2* [14].

## 3. Results

### 3.1. Composition of Albumen and Yolk

Table 1 reports the least-squares means of the composition of the albumen. PP and RL had the greatest protein wet basis content (9.73 ± 0.71% and 9.72 ± 0.61%, respectively), and differed from PA-C, PO-B and PO-W breeds (*p* < 0.05). On the contrary, PP and RL had the lowest moisture content (88.63 ± 0.88% and 88.60 ± 0.78%), and differed from PA-C, PO-B, PO-W and RM (*p* < 0.05), with the greatest moisture observed for PO-B (90.06 ± 1.23%). As regards ash content, the only significant difference was observed between ER (0.72 ± 0.06%) and PO-B (0.67 ± 0.06%; *p* < 0.05).

According to the yolk wet basis composition (Table 2), eggs of PA-C (16.55 ± 0.34%) and PO-W (16.57 ± 0.31%) had the greatest protein content, whereas eggs of RL (15.21 ± 0.63%) and RM (15.62 ± 1.13%) had the lowest protein content (*p* < 0.05). Moisture content was greater in eggs of PA-C and PP (50.11 ± 0.83% and 50.08 ± 1.04%, respectively) than ER (48.64 ± 0.85%), PO-B (48.86 ± 0.74%), PO-W (48.98 ± 0.72%), RL (48.55 ± 0.58%) and RM (49.14 ± 0.84%) (*p* < 0.05). The greatest lipid wet basis content was for RL (35.30 ± 0.98%) and it differed from the lipid content of all the other breeds (*p* < 0.05), with the lowest content observed for PA-C and PP (32.14 ± 0.93% and 32.73 ± 1.11%, respectively). Nevertheless, cholesterol content on a wet basis was the greatest in PP yolk (1406.39 ± 82.34 mg/100 g) and it differed from that of PA-C (1243.81 ± 134.13 mg/100 g), PO-W (1227.08 ± 71.14 mg/100 g) and RM (1262.27 ± 84.12 mg/100 g) (*p* < 0.05). The greatest wet basis ash content was observed in PA-G (1.89 ± 0.21%) and the lowest were seen in PO-B (1.67 ± 0.24%), PO-W (1.71 ± 0.21%) and RL (1.71 ± 0.15%) (*p* < 0.05). The same differences were observed for dry matter.

### 3.2. Differences in Chemical Composition between White and Tinted Eggshell Breeds

Differences in composition between white eggshell (PA-C, PA-G, PO-W, PO-B) and tinted eggshell breeds (ER, RM, RL, PP) were investigated. Figure 2 depicts boxplots with the comparison of protein, moisture, dry matter and ash of albumen between the two groups. The violin plot has been integrated into boxplots to show the data distribution. Protein content and dry matter were greater in eggs with tinted shells (*p* < 0.001), whereas moisture content was greater in eggs with white shells (*p* < 0.001). Ash content did not differ between the two groups (Figure 2).

Figure 3 depicts the boxplots of the yolk content of eggs with white and tinted shells. In particular, lipid content (*p* < 0.001) and cholesterol content (*p* < 0.01) were greater in eggs with tinted than white shells. Conversely, the protein content was significantly greater in eggs with white than tinted shells. Moisture, dry matter and ash did not differ between the two groups (Figure 3).

### 3.3. Prediction Models

Independent calibrations were performed for the fresh and freeze-dried eggs. The performance statistics of the best VIS-NIR prediction of fresh and freeze-dried albumen and yolk samples are reported in Table 3 and Table 4, respectively. The number of terms ranged from 5 to 9 for albumen and 1 to 9 for yolk. In general, among the calibration models tested, the optimal prediction equations were mainly composed by NONE scatter, SNV_D and MSC, and half of the best equations were developed using the first derivative with a gap of four data points.

In general, the best performance of prediction was obtained for the freeze-dried product in which the RPD_CV_ for dry matter in albumen was 4.47, and in which it was 3.48 and 2.68 for protein and dry matter in yolk, respectively. On the other hand, RPD_CV_ lower than 2 was reported for all composition traits in the fresh yolk, and also for ash (% DM), lipids and cholesterol in the freeze-dried yolk. In albumen, all traits reported RPD_CV_ lower than 2, with the exception of the dry matter in freeze-dried albumen.

## 4. Discussion

### 4.1. Chemical Composition

The importance of the genotype (hybrids, local breeds) to explain the variation of egg composition has been reported in the literature [15]. The characterization and valorization of local breeds through the qualitative aspects of the product is a viable strategy to protect and promote poultry biodiversity. The amount of protein in albumen reported for PA-G, PP and RL in the present study was greater than that observed in the literature for commercial hybrids [16,17]. In the current trial, proteins of the albumen varied among breeds, being 12% greater in PP and RL than in PA-C, PO-B and PO-W. For the albumen ash, differences among the breeds were about 7% (ER vs. PO-B). For the yolk, changes of about 7–8% were observed for the protein and lipid contents between some white and tinted eggshell breeds: eggs of PO-W and PA-C (white eggshell breeds) had greater protein content than eggs of RL and RM (tinted eggshell breeds), and the contrary was true for the lipids (RL vs. PO-W). Lipid content of RL eggs was greater also than in PP, a tinted eggshell breed, but these eggs also had very high shell lightness. The greatest variations among the breeds were observed for the yolk ash (13%) and cholesterol content (12%): ash varied between PA and PO, and cholesterol between PP and PA-C, but it differed also from RM. In detail, the content of protein, ash and dry matter in the eggs of the investigated local breeds was similar to that observed in two Italian native breeds of the Emilia Romagna region (Modenese and Romagnolo) [17]. Moreover, Zanon et al. [17] reported that local breeds produced eggs with greater protein content (+5.7%) and lower ether extract content (-5.3%) in yolk than the commercial white and tinted eggshell hybrids. Ianni et al. [1] compared the cholesterol and lipid content of the yolk between the Nera Atriana local breed and a commercial hybrid, and the differences were not significant. However, the results of the present study highlighted that PP had greater cholesterol and lower lipid content than those observed for commercial hybrids in the study of Ianni et al. [1].

In the study of Lordelo et al. [18], eggs from hybrid hens had greater albumen and lower yolk percentage when compared to eggs from local hens. The percentage of albumen was greater also in eggs of commercial hybrids than local hens in Rizzi and Marangon [19]. Overall, differences between eggs of local breeds and hybrids are often related to the albumen [17,18,19,20,21]. In particular, the water content is greater in the albumen of hybrid hens, likely as a consequence of the genetic improvement in the weight of the egg, which has led to an increase in the water content of the albumen [19]. A comparison, conducted between the composition of eggs from the current trial with that of hybrid hens with similar egg size, supports there being a greater amount of water in the albumen of commercial eggs [19]. Overall, comparisons among studies should be performed with caution, as egg composition varies according to the breed, its laying rate, feeding and, in general, environmental conditions.

The different eggshell color of the eggs laid by the hens, as influenced by their different genetic origin, also reflects different body size and metabolism, which in turn affect the egg formation and production. The cholesterol content may be affected by the laying rate, as reported in the literature [22]. At the age of egg sampling, the PP hens showed a laying rate lower than that of the other breeds, which exhibited a dilution effect for the cholesterol content. Differences in egg composition (moisture, fat and protein contents) observed across the local breeds are, at least partly, related to the genetic background of the hens [2]. Whereas, as expected, similarities were observed between Polverara and Padovana breeds due to their close genetic background [2].

Overall, the composition of both albumen and yolk differed among breeds. Considering the eggshell color, the expected difference in the composition of the eggs has been already demonstrated in previous studies on various species [23,24]. The chemical outcome seems to underline a better nutritional profile of eggs with tinted shells in terms of greater protein content in albumen. However, the yolk had greater amount of lipids and cholesterol than the yolk of white-shell eggs. The color of the eggshell is controlled by several genes that encode proteins and enzymes, thereby regulating the production and deposition of pigment into the shell [25]. Thus, the differences can be explained by the genetics of the breeds. Samiullah et al. (2015) [25] reported that eggshell and egg internal quality are influenced by various factors such as egg weight, shell weight, specific gravity, shell breaking strength, shell deformation, shell thickness, albumen height and yolk color. However, tinted eggshell color is positively correlated with some shell characteristics, such as shell strength and hatchability [26]. Further, it has been suggested that some shell quality features such as strength, weight, consistency and ultrastructure can be assessed via shell color because of significant correlations between the shell quality index and shell color [27,28]. However, other authors have reported inconsistent results in this regard [29,30], and therefore shell color cannot be applied reliably as a quality assessment tool [26].

### 4.2. VIS-NIR Prediction Models

The ability of VIS-NIR to predict the composition of eggs has been evaluated in commercial hens [31]. However, to the best of our knowledge, no previous studies have assessed the prediction capability of VIS-NIR for the composition of eggs from local chicken breeds. The best performance of prediction was observed for freeze-dried yolk and albumen (Table 3 and Table 4, respectively), which was likely due to the instability of the moisture in the fresh samples exposed to the ambient air during the spectra acquisition. Moreover, the divergences among the trait prediction between fresh and freeze-dried samples could be mostly attributed to a scattering effect due to the nature of the product [32]. In particular, the presence of high amounts of water in food matrices masks the peaks of the spectrum of the other chemical components; whereas, low water content allows us to detect the peaks of the functional groups easily [33]. Therefore, the performance of the prediction models on dehydrated samples improved.

In fresh yolk, the performance of prediction for protein (R^2^_CV_ = 0.68) and for lipids (R^2^_CV_ = 0.59) indicate an approximative quantification and a discriminative capability, respectively [34,35,36]. Considering the performance observed for cholesterol, and as reported in the study of Dalle Zotte et al. (2006) [36], the unpredictability of this trait was expected due to the low concentration (~0.02%) close to the limit of sensitivity (<0.1%) of infrared spectroscopy [36]. Similarly, the performance of predictions in fresh albumen (Table 3) which were observed for dry matter (R^2^_CV_ = 0.73) and protein (R^2^_CV_ = 0.66) suggest an approximative quantification of the models. More satisfying predictions were obtained in freeze-dried albumen and yolk, except for protein, due to the lower amount of water. In detail, dry matter and protein models in the freeze-dried yolk are considered good and excellent for quality control and for practical purposes, respectively [35]. Conversely, the lipid calibration indicated an approximative quantification of such a trait. Moreover, in freeze-dried albumen an excellent prediction was obtained for dry matter, and thus such a prediction model can be used for quantitative purposes in the substitution of chemical laboratory analysis. The overall incapability of VIS-NIR to predict ash content is due to the non-interaction between VIS-NIR radiation and inorganic compounds, as previously reported in eggs and other food products [36,37,38,39]. Similar to our study, Dalle Zotte et al. (2006) [36] reported that lipid and cholesterol contents of freeze-dried yolk were better predicted using SNV_D scatter correction as the optimum pretreatment. Some discrepancies were observed in the mathematical equations selected for the prediction of protein, ash and dry matter content between the current study and that of Dalle Zotte et al. (2006) [36]. Overall, the present study reported better prediction performance for dry matter and protein in freeze-dried yolk compared to the aforementioned study. Divergences in protein could be related to the indirect determination method, i.e., its calculation as difference from moisture, ash and lipids, compared to the direct quantification methods of the present study. Zhao et al. (2018) [31] proposed independent prediction equations for yolk and albumen, and they reported worse performances in external validation for protein (R^2^ = 0.81) and moisture content (R^2^ = 0.76), which was probably due to the different type of validation approach used [40].

## 5. Conclusions

The present study has characterized the composition of the eggs from eight local chicken breeds of the Veneto region and has developed calibration models for its prediction. The composition of the egg is certainly important to valorize autochthonous chicken breeds and contribute to their conservation. The VIS-NIR models, developed in the current study, can be helpful for this purpose. Indeed, they can provide fast, relatively cheap and, overall, satisfactory predictions for composition traits, thus allowing us to discriminate between high and low trait concentrations in the egg, and thus for breed traceability. Moreover, calibration models may provide a useful tool to discriminate between commercial and local products according to their composition. Therefore, future studies should consider commercial lines to improve fitting statistics of the prediction models and investigate the ability of the models to discriminate between commercial and local chicken breeds according to the composition of their eggs.

## Figures and Tables

**Figure 1 animals-13-00077-f001:**
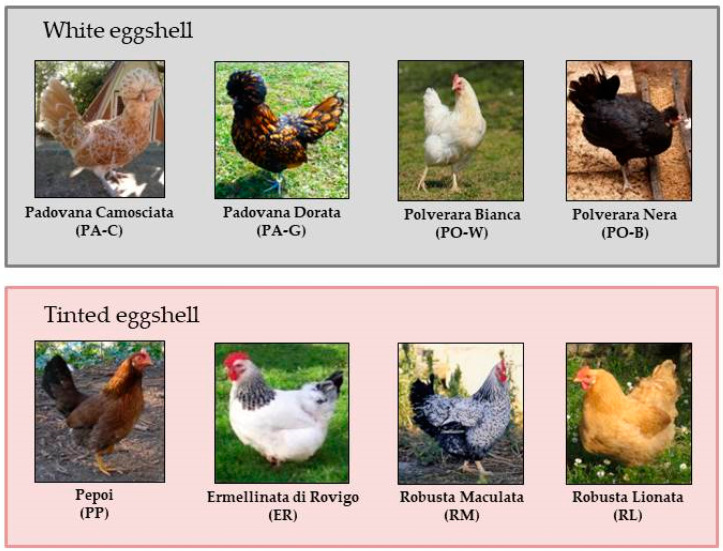
Local chicken breeds involved in the trial grouped according to eggshell color.

**Figure 2 animals-13-00077-f002:**
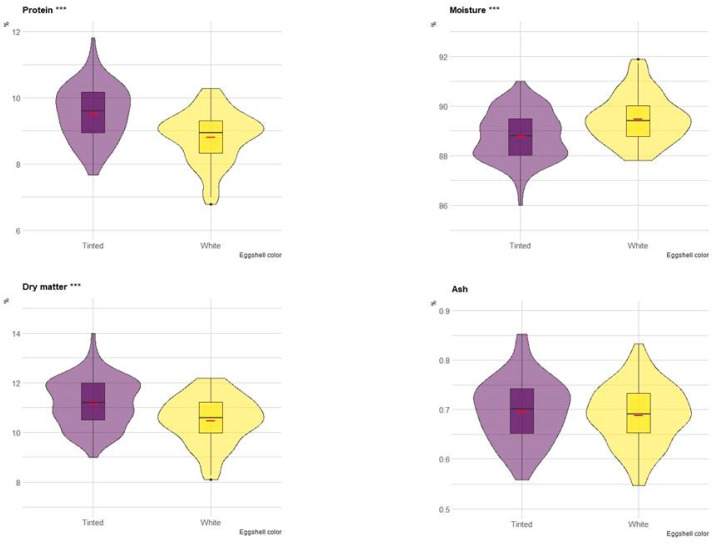
Boxplot of the composition (on a wet basis) of albumen in eggs of tinted (violet) and white (yellow) shell. Red dash indicates mean value. The violin blot represents the data distribution. *** *p* < 0.001.

**Figure 3 animals-13-00077-f003:**
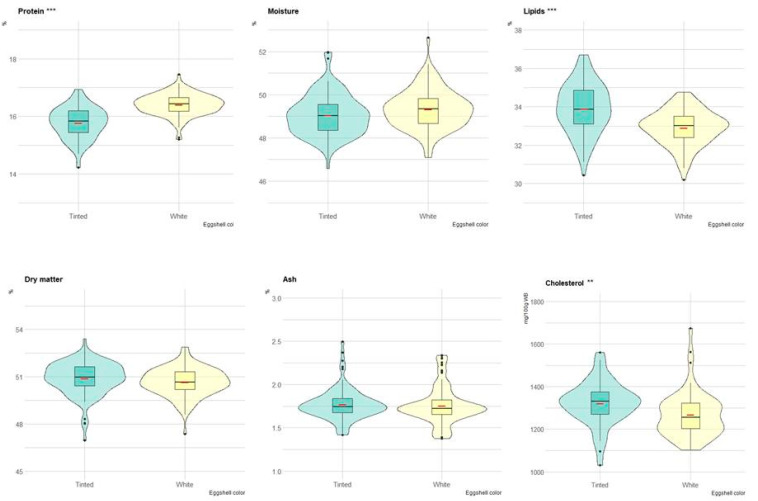
Boxplot of the composition (on a wet basis) of yolk in eggs of tinted (violet) and white (yellow) shell. Red dash indicates mean value. The violin blot represents the data distribution. ** *p* < 0.01; *** *p* < 0.001.

**Table 1 animals-13-00077-t001:** Least-squares means and standard error of composition of albumen for the breed effect.

Compound	Breed							
	PA-C	PA-G	PO-W	PO-B	PP	ER	RM	RL
Moisture (% WB)	89.48 ± 0.97 ^ab^	88.92 ± 0.68 ^bc^	89.57 ± 1.31 ^ab^	90.06 ± 1.23 ^a^	88.63 ± 0.88 ^c^	89.08 ± 1.06 ^bc^	88.90 ± 1.12 ^b^	88.60 ± 0.78 ^c^
Protein (% WB)	8.81 ± 0.71 ^bc^	9.33 ± 0.48 ^ab^	8.81 ± 1.27 ^bc^	8.34 ± 1.27 ^c^	9.73 ± 0.71 ^a^	9.25 ± 0.98 ^ab^	9.44 ± 0.91 ^ab^	9.72 ± 0.61 ^a^
Ash (% WB)	0.69 ± 0.07 ^ab^	0.71 ± 0.04 ^ab^	0.69 ± 0.06 ^ab^	0.67 ± 0.06 ^b^	0.68 ± 0.06 ^ab^	0.72 ± 0.06 ^a^	0.69 ± 0.07 ^ab^	0.69 ± 0.05 ^ab^
Protein (% DM)	83.88 ± 1.99 ^bc^	84.24 ± 1.73 ^ab^	84.39 ± 1.45 ^bc^	83.94 ± 1.46 ^c^	85.61 ± 1.29 ^a^	84.72 ± 2.02 ^ab^	85.09 ± 1.45 ^ab^	88.91 ± 0.37 ^a^
Ash (% DM)	6.57 ± 0.37 ^ab^	6.33 ± 0.32 ^ab^	6.68 ± 0.49 ^ab^	6.75 ± 0.38 ^b^	6.00 ± 0.39 ^ab^	6.70 ± 0.49 ^a^	6.24 ± 0.32 ^ab^	6.03 ± 0.36 ^ab^

Abbreviations: WB, wet basis; DM, dry matter; PA-C, Padovana Camosciata; PA-G, Padovana Dorata; PO-W, Polverara Bianca; PO-B, Polverara Nera; PP, Pepoi; ER, Ermellinata di Rovigo; RM, Robusta Maculata; RL, Robusta Lionata. ^a,b,c^ Means with different letters within a row differ significantly (*p* < 0.05).

**Table 2 animals-13-00077-t002:** Least-squares means and standard error of composition of yolk for the breed effect.

	Breed							
	PA-C	PA-G	PO-W	PO-B	PP	ER	RM	RL
Moisture (% WB)	50.11 ± 0.83 ^a^	49.40 ± 0.83 ^ab^	48.98 ± 0.72 ^bc^	48.86 ± 0.74 ^bc^	50.08 ± 1.04 ^a^	48.64 ± 0.85 ^c^	49.14 ± 0.84 ^bc^	48.55 ± 0.58 ^c^
Protein (% WB)	16.55 ± 0.34 ^a^	16.35 ± 0.35 ^ab^	16.57 ± 0.31 ^a^	16.18 ± 0.37 ^b^	16.08 ± 0.41 ^b^	16.16 ± 0.35 ^b^	15.62 ± 1.13 ^c^	15.21 ± 0.63 ^d^
Lipid (% WB)	32.14 ± 0.93 ^f^	33.02 ± 0.66^cde^	32.83 ± 0.66^def^	33.64 ± 0.79 ^bc^	32.73 ± 1.11^ef^	33.54 ± 0.85 ^bcd^	34.11 ± 1.12 ^b^	35.30 ± 0.98 ^a^
Ash (% WB)	1.77 ± 0.12 ^abc^	1.89 ± 0.21 ^a^	1.71 ± 0.21 ^bc^	1.67 ± 0.24^c^	1.84 ± 0.22 ^ab^	1.74 ± 0.21 ^abc^	1.78 ± 0.09 ^abc^	1.71 ± 0.15 ^bc^
Cholesterol (mg/100 g WB)	1243.81 ± 134.13 ^b^	1284 ± 105.79 ^ab^	1227.08 ± 71.14 ^b^	1317.71 ± 89.89 ^ab^	1406.39 ± 82.34 ^a^	1319.77 ± 143.37 ^ab^	1262.27 ± 84.12 ^b^	1308.72 ± 57.24 ^ab^
Protein (% DM)	33.17 ± 0.45 ^a^	32.30 ± 0.68 ^ab^	32.48 ± 0.50 ^a^	31.65 ± 0.63 ^b^	32.23 ± 077 ^b^	31.47 ± 0.60 ^b^	30.70 ± 0.68 ^c^	29.56 ± 0.77 ^d^
Lipid (% DM)	64.41 ± 1.00 ^f^	65.23 ± 0.95 ^cde^	64.36 ± 1.03 ^def^	65.77 ± 1.17 ^bc^	65.66 ± 1.68 ^ef^	65.30 ± 0.89 ^bcd^	67.05 ± 1.34 ^b^	68.60 ± 1.34 ^a^
Ash (% DM)	3.54 ± 0.24 ^abc^	3.73 ± 0.42 ^a^	3.35 ± 0.38 ^bc^	3.27 ± 0.47 ^c^	3.69 ± 0.41 ^ab^	3.39 ± 0.38 ^abc^	3.51 ± 0.16 ^abc^	3.33 ± 0.29 ^bc^
Cholesterol (mg/100 g DM)	2486.82 ± 243.33 ^b^	2545.29 ± 200.97 ^ab^	2410.80 ± 156.93 ^b^	2577.53 ± 177.32 ^ab^	2807.85 ± 124.47 ^a^	2562.90 ± 277.19 ^ab^	2478.30 ± 170.47 ^b^	2546.45 ± 95.38 ^ab^

Abbreviations: WB, wet basis; DM, dry matter; PA-C, Padovana Camosciata; PA-G, Padovana Dorata; PO-W, Polverara Bianca; PO-B, Polverara Nera; PP, Pepoi; ER, Ermellinata di Rovigo; RM, Robusta Maculata; RL, Robusta Lionata. ^a,b,c,d,e,f^ Means with different letters within a row differ significantly (*p* < 0.05).

**Table 3 animals-13-00077-t003:** Calibration and cross-validation statistics for modified partial least-squares regression models developed to predict the composition of fresh and freeze-dried albumen.

	Math		T	n	Mean	SD	SE_C_	R^2^_C_	SE_cv_	R^2^_cv_	RPD_cv_
*Fresh Albumen*											
Dry matter (%)	SNV_D	1441	9	168	11.07	1.08	0.48	0.81	0.57	0.73	1.91
Protein (%)	NONE	2551	5	167	9.32	0.87	0.43	0.75	0.51	0.66	1.70
Ash (%)	NONE	1441	6	170	0.70	0.06	0.04	0.40	0.05	0.26	1.16
*Freeze-Dried Albumen*											
Dry matter	MSC	1441	5	176	89.11	1.58	0.33	0.96	0.35	0.95	4.47
Protein (% DM)	NONE	1441	6	170	84.65	1.71	1.10	0.58	1.18	0.52	1.45
Ash (% DM)	MSC	0011	9	176	6.33	0.48	0.28	0.66	0.30	0.60	1.58

Abbreviations: Math, mathematical treatment; MSC, multiplicative scatter correction; SNV_D, standard normal variate and detrend; NONE, no correction; T, number of terms; n, number of samples used to develop the model; SD, standard deviation of reference data; SE_C_, standard error of calibration; R^2^_C_, coefficient of determination of calibration; SE_CV_, standard error of cross-validation; R^2^_CV_, coefficient of determination of cross-validation; RPDcv, residual predictive deviation of cross-validation calculated as the ratio between SD and SEcv.

**Table 4 animals-13-00077-t004:** Calibration and cross-validation statistics for modified partial least-squares regression models developed to predict the composition of fresh and freeze-dried yolk.

	Math		T	n	Mean	SD	SE_C_	R^2^_C_	SE_cv_	R^2^_cv_	RPD_cv_
*Fresh Yolk*											
Dry matter (%)	NONE	2551	3	186	50.93	0.81	0.60	0.44	0.65	0.36	1.25
Protein (%)	NONE	1441	9	190	16.22	0.55	0.29	0.72	0.31	0.68	1.75
Lipids (%)	SNV_D	2551	5	190	33.22	1.14	0.62	0.70	0.73	0.59	1.56
Ash (%)	NONE	0011	2	176	1.73	0.14	0.13	0.14	0.13	0.13	1.07
Cholesterol (mg/100 g)	MSC	2551	1	100	1256.77	92.56	81.37	0.23	84.47	0.16	1.10
*Freeze-Dried Yolk*											
Dry matter	NONE	2551	8	186	99.06	0.80	0.24	0.91	0.30	0.86	2.68
Protein (% DM)	NONE	1441	7	190	31.92	1.10	0.30	0.93	0.32	0.92	3.48
Lipids (% DM)	SNV_D	1441	9	194	65.29	1.75	0.83	0.78	0.89	0.74	1.98
Ash (% DM)	SNV_D	0011	1	182	3.38	0.28	0.26	0.14	0.27	0.12	1.06
Cholesterol (mg/100 g DM)	SNV_D	1441	4	127	2477.37	177.45	101.29	0.67	129.14	0.48	1.37

Abbreviations: Math, mathematical treatment; MSC, multiplicative scatter correction; SNV_D, standard normal variate and detrend; NONE, no correction; T, number of terms; n, number of samples used to develop the model; SD, standard deviation of reference data; SE_C_, standard error of calibration; R^2^_C_, coefficient of determination of calibration; SEcv, standard error of cross-validation; R^2^_CV_, coefficient of determination of cross-validation; RPDcv, residual predictive deviation of cross-validation calculated as the ratio between SD and SEcv.

## Data Availability

Not applicable.

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
