# Peer review of "Egg Quality of Italian Local Chicken Breeds: II. Composition and Predictive Ability of VIS-Near-InfraRed Spectroscopy"

_animals, 2022, doi:10.3390/ani13010077_

Round 1
Reviewer 1 Report
This manuscript tried to develop a VIS-NIR model to predict the chemical composition of eggs from local chicken breeds, which arose my interest, and it will provide a good way to predict the egg quality. The egg quality was affected by many factors, including breeds, ages, nutritional levels, environmental condition, storage time, etc., some more information about the egg samples need to be supplemented in this study. The detailed comments are as follows:
1. Introduction, second paragraph: “hens eggs” should be “eggs”;
2. 2.1 Hens: there had 8 local breeds, how many hens for each breed? How about the ages for the breeds? Are these birds were reared at the same time? Some more information about the feeding and management had better be supplemented.
3. 2.2, about the 200 samples, when do the eggs were sampled? “one day eggs” mean eggs from the same day? How about the hen age? Different age will affect the egg quality.
4. Pelleted feed (chemical composition), should be “composition”; what is the nutritional level based on?
5. 2.3 “Individual spectra were collected on the albumen and the yolk separately, both on fresh and freeze-dried samples.”, while in 2.2, the chemical composition of albumen and yolk were analyzed after freeze-drying, which may partly explain the result that the prediction models performed better for freeze-dried sample than fresh albumen and yolk.
6. 3.1: what is “wet basis content”, how to calculate it?According to 2.2, the albumen and yolk are taken freeze-dried samples.
7. Figure 1 and 2: I think the data is clearer than the boxplot.
8. Discussion: 4.1, the first and the second sentence are repetitive; why not directly mentioned?
9. “in the current trial, the hens aged 50 weeks, but they showed different laying rate” this age information should be stated in the Material and Method.
10. 4.2: for the first sentence, the relevant references should be supplemented.
11. Explain why the prediction models performed better for freeze-dried sample than fresh albumen and yolk.
12. Pay attention to the expressions, e.g., “the animals” had better be “the birds” or “the hens”; “per each breed” be “each breed”; “egg chemical composition” be “chemical composition of eggs”; “The prediction models performances”be “the performance of the prediction model”, etc.
Author Response
Dear Editor,
We have put all our efforts to address as better as possible the points raised by the Reviewers. Please find below a detailed reply to the point-by-point issues raised by the Reviewers. Changes in the text are in red to make Reviewers’ job easier. We would like to thank the Reviewers for their time and thorough evaluation of our manuscript. We hope that our revisions/answers meet with your approval.
Thank you very much in advance.
Sincerely,
Sarah Currò (on behalf of all authors)
#Reviewer 1
- Introduction, second paragraph: “hens eggs” should be “eggs”;
AU: Thank you. Changed as suggested.
- 2.1 Hens: there had 8 local breeds, how many hens for each breed? How about the ages for the breeds? Are these birds were reared at the same time? Some more information about the feeding and management had better be supplemented.
AU: For this study, 60 hens per breed were available. Hens averaged 50 weeks of age (with very small variation, from 49 to 51 weeks of age). Fifty eggs per breed across 4 consecutive days (about 13 eggs/breed/day) were collected. This resulted in a total of 400 eggs collected during the trial and 200 final samples used for composition analyses, since each sample was composed of two eggs. These details have been added to M&M now.
- 2.2, about the 200 samples, when do the eggs were sampled? “one day eggs” mean eggs from the same day? How about the hen age? Different age will affect the egg quality.
AU: Eggs were collected every day during 4 consecutive days. “one day eggs” means that eggs were collected within 24 hours from laying. As regards age, please, see answer to your previous comment. Because of the very narrow range of age (from 49 to 51 weeks), it is very unlikely that age has had an effect on egg quality. The description of hens age has been moved to M&M as suggested by Reviewer 2.
- Pelleted feed (chemical composition), should be “composition”; what is the nutritional level based on?
AU: The sentence has been modified and information on the nutritional level added to the manuscript.
- 2.3 “Individual spectra were collected on the albumen and the yolk separately, both on fresh and freeze-dried samples.”, while in 2.2, the chemical composition of albumen and yolk were analyzed after freeze-drying, which may partly explain the result that the prediction models performed better for freeze-dried sample than fresh albumen and yolk.
AU: Several papers have reported the chemical analysis on the freeze-drying samples to optimize the vis-NIRs predictions, e.g.:
- Franks, F. (1998). Freeze-drying of bioproducts: putting principles into practice. European journal of Pharmaceutics and BioPharmaceutics, 45, 221-229, https://doi.org/10.1016/S0939-6411(98)00004-6
- Giaretta, E., Mordenti, A., Palmonari, A., Brogna, N., Canestrari, G., Belloni, P., ... & Formigoni, A. (2019). NIRs calibration models for chemical composition and fatty acid families of raw and freeze-dried beef: A comparison. Journal of Food Composition and Analysis, 83, 103257, https://doi.org/10.1016/j.jfca.2019.103257
- Viljoen, M., Hoffman, L. C., & Brand, T. S. (2005). Prediction of the chemical composition of freeze dried ostrich meat with near infrared reflectance spectroscopy. Meat Science, 69, 255-261, https://doi.org/10.1016/j.meatsci.2004.07.008.
Indeed, the prediction models perform better for freeze-dried samples than fresh albumen and yolk because the capability of Vis-NIRs methodology is inversely proportional to the water content (the presence of water in a sample can mask the presence of relevant peaks in the spectrum).
- 3.1: what is “wet basis content”, how to calculate it?According to 2.2, the albumen and yolk are taken freeze-dried samples.
AU: According to the official analytical methods, egg samples underwent to the freeze-dried process to perform the chemical analysis. However, moisture content was determined by a thermogravimetric approach recording the moisture evaporation amount and obtaining the relative dry matter as: 100 - moisture content. Thus, the analytes expressed on dry matter were converted on wet basis through the multiplication by the dry matter content divided 100.
- Figure 1 and 2: I think the data is clearer than the boxplot.
AU: Thank you for the comment. The boxplot and the violin blot were used because we believe that they can easily represent together the descriptive statistics and the distribution of data, respectively.
- Discussion: 4.1, the first and the second sentence are repetitive; why not directly mentioned?
AU: The sentences have been modified.
- “in the current trial, the hens aged 50 weeks, but they showed different laying rate” this age information should be stated in the Material and Method.
AU: The sentence has been moved to M&M.
- 4.2: for the first sentence, the relevant references should be supplemented.
AU: A reference has been added.
- Explain why the prediction models performed better for freeze-dried sample than fresh albumen and yolk.
AU: The following sentence has been added to the manuscript to address this issue: “In particular, foods with high water content show spectra peaks similar to pure water, whereas foods with low water content show a different pattern due to the uncovered peaks of the functional groups of other components. Thus, water affects the frequencies and intensity of bands derived by variation of the strength of hydrogen bonds and hydration level. As a consequence, when prediction models are performed on dehydrated samples the performance of chemical traits are generally improved.” (Büning-Pfaue, H. 2003. Analysis of water in food by near infrared spectroscopy. Food Chemistry, 82, 107-115, https://doi.org/10.1016/S0308-8146(02)00583-6).
- Pay attention to the expressions, e.g., “the animals” had better be “the birds” or “the hens”; “per each breed” be “each breed”; “egg chemical composition” be “chemical composition of eggs”; “The prediction models performances”be “the performance of the prediction model”, etc.
AU: Thank you for this comment. The manuscript has been adjusted accordingly.

Reviewer 2 Report
The study deals with a comparison of the chemical egg quality of Italian chicken breeds. The manuscript is well structured and easy to follow. However, particularly in the material and methods section, some additions and corrections seem necessary to meet scientific criteria.
A general flaw of the study is the lack of a control group with high performing hybrid laying hen strains. Thus, while a comparison between chicken breeds is possible, no direct statement can be made how egg quality compares to the standard of agricultural production. This should be critically examined, at least in the discussion of methods.
References:
The number of references appears to be low, as there are many publications of high evidence level on the performance and product traits of purebred chickens. This should be corrected. In the introduction, there should also be a link to the performance of local chickens (rearing/fattening and laying performance), as these are the crucial criteria why they are not used commercially.
Abstract:
Please add the p-values for your major outcomes.
M&M
Chicken breeds undergo a substantial exterieur selection, but not all readers may be familiar with the phenotype of the breeds. Therefore, fotographs of the breeds should be inserted as figures. Moreover, a brief characterization of the breeds regarding body mass color should be included.
The description of the housing conditions is insufficient: barn size, stocking density, litter, utilities, free-range area, light regime, etc. must be accurately presented here.
For all devices/materials used in the study, please consistently list the specific product name and manufacturer including location and country.
It is recommended to check and describe much more precisely the sampling of eggs: How was ensured that the sample size of 25 eggs per breed is sufficient to address the scientific questions? Was the sample size calculated as part of the study design? At what animal age, how many eggs were collected? Are all stages of the laying period represented? The age of the survey time points must be added.
Statistical analyses:
Overall, the analyses should be presented more concisely. In addition, references recommending the procedures should be added.
Among others, information is missing on: statistical software, testing for normal distribution, correction for multiple testing.
Discussion:
A critical comparison to chemical properties of eggs from hybrid strains should be added.
Author Response
Dear Editor,
We have put all our efforts to address as better as possible the points raised by the Reviewers. Please find below a detailed reply to the point-by-point issues raised by the Reviewers. Changes in the text are in red to make Reviewers’ job easier. We would like to thank the Reviewers for their time and thorough evaluation of our manuscript. We hope that our revisions/answers meet with your approval.
Thank you very much in advance.
Sincerely,
Sarah Currò (on behalf of all authors)
#Reviewer 2
The study deals with a comparison of the chemical egg quality of Italian chicken breeds. The manuscript is well structured and easy to follow. However, particularly in the material and methods section, some additions and corrections seem necessary to meet scientific criteria.
AU: We would like to thank the Reviewer for the general favorable comments and suggestions.
A general flaw of the study is the lack of a control group with high performing hybrid laying hen strains. Thus, while a comparison between chicken breeds is possible, no direct statement can be made how egg quality compares to the standard of agricultural production. This should be critically examined, at least in the discussion of methods.
AU: Thank you for the suggestion. Unfortunately, as you correctly pointed out, we did not have a control group in our trial. The conservation center only has local breeds, some of which at very high risk of extinction, with the purpose of safeguarding them. Commercial lines are not admitted to the center. We explored the opportunity/idea to look for external farms and consider hybrid laying hens as control group, but the option was discarded because having those animals under completely different environmental and management conditions (e.g., diet) could have introduced a bias in our trial. Moreover, having hybrids and local breeds in different farms means that the effect of genotype is nested within breed (hierarchical model) and this makes difficult to disentangle the genotype and farm effects. Nevertheless, to address Reviewer’s comment, we have put more efforts to improve the discussion through studies that dealt with characteristics of commercial lines, and to compare these with local breeds (some discussion was already present in the original manuscript).
References:
The number of references appears to be low, as there are many publications of high evidence level on the performance and product traits of purebred chickens. This should be corrected. In the introduction, there should also be a link to the performance of local chickens (rearing/fattening and laying performance), as these are the crucial criteria why they are not used commercially.
AU: The Introduction has been improved as well as the discussion according to Reviewer’s comment. We would like to thank the Reviewer for this suggestion.
Abstract:
Please add the p-values for your major outcomes.
AU: All relevant p-values for the major outcomes were already present in the original submitted manuscript.
M&M
Chicken breeds undergo a substantial exterieur selection, but not all readers may be familiar with the phenotype of the breeds. Therefore, fotographs of the breeds should be inserted as figures. Moreover, a brief characterization of the breeds regarding body mass color should be included.
AU: We would like to thank the Reviewer for this suggestion. A new figure presenting pictures of the breeds involved in the study (Figure 1) has been added to the manuscript (M&M). About “body mass color”, this description looks marginal in the current manuscript and, if the Reviewer means plumage color, it can be inferred from the pictures.
The description of the housing conditions is insufficient: barn size, stocking density, litter, utilities, free-range area, light regime, etc. must be accurately presented here.
AU: Thank you for the comment. More details have been added to M&M.
For all devices/materials used in the study, please consistently list the specific product name and manufacturer including location and country.
AU: Added as suggested (see M&M).
It is recommended to check and describe much more precisely the sampling of eggs: How was ensured that the sample size of 25 eggs per breed is sufficient to address the scientific questions? Was the sample size calculated as part of the study design? At what animal age, how many eggs were collected? Are all stages of the laying period represented? The age of the survey time points must be added
AU: Thank you for the comments. For this study, 60 hens per breed were available. Hens averaged 50 weeks of age, with very small variation, from 49 to 51 weeks of age (the information on average age at sampling was already included in the original submission, but it has been moved from the Discussion to M&M). Fifty eggs per breed across 4 consecutive days (10 eggs/breed/day) were collected. This resulted in a total of 400 eggs collected during the trial and 200 final samples used for composition analyses, since each sample was composed of two eggs. These details have been added to M&M now, also following Reviewer’s 1 comments. Unfortunately, the number of eggs and samples was related to the low oviposition of the breeds (we would reinforce that breeds involved in the current study undergo a conservation program, but it is not easy to keep the inbreeding at acceptable values due to the quite low number of individuals/breed, and thus to avoid negative impact on reproductive traits). Also, looking at the scientific literature, several studies considered from 17 to 50 eggs for similar studies on local breeds, e.g.:
- Zanon, A., Beretti, V., Superchi, P., Zambini, E. M., & Sabbioni, A. (2006). Physico-chemical characteristics of eggs from two Italian autochthonous chicken breeds: Modenese and Romagnolo. World's Poultry Science Journal, 62(Suppl), 203.
- Van Duy, N., Moula, N., Do Duc, L., Pham Kim, D., Dao Thi, H., Bui Huu, D., ... & Farnir, F. (2015). Ho chicken in Bac Ninh Province (Vietnam): From an indigenous chicken to local poultry breed. International Journal of Poultry Science, 14(9), https://dx.doi.org/10.3923/ijps.2015.521.528 - Ariza, A. G., González, F. J. N., Arbulu, A. A., Bermejo, J. V. D., & Vallejo, M. E. C. (2021). Hen breed and variety factors as a source of variability for the chemical composition of eggs. Journal of Food Composition and Analysis, 95, 103673, https://doi.org/10.1016/j.jfca.2020.103673.
Statistical analyses:
Overall, the analyses should be presented more concisely. In addition, references recommending the procedures should be added.
AU: Thank you. We would like to point out that statistical analyses have been already presented concisely. We would avoid further shortening as this would result in loss of relevant information for the reader.
Among others, information is missing on: statistical software, testing for normal distribution, correction for multiple testing.
AU: Information on statistical software and testing for normal distribution has been included now. Information on multiple testing (Tukey’s multiple comparison test) was already present in the original submission (chapter 2.5).
Discussion:
A critical comparison to chemical properties of eggs from hybrid strains should be added.
AU: Thank you for the suggestion. See our answer to your first comment related to the flaw of our study.

Round 2
Reviewer 2 Report
The manuscript can be accepted as submitted.